# Variability of *ACOX1* Gene Polymorphisms across Different Horse Breeds with Regard to Selection Pressure

**DOI:** 10.3390/ani10122225

**Published:** 2020-11-27

**Authors:** Grzegorz Myćka, Adrianna D. Musiał, Monika Stefaniuk-Szmukier, Katarzyna Piórkowska, Katarzyna Ropka-Molik

**Affiliations:** 1Faculty of Biotechnology and Horticulture, University of Agriculture in Krakow, al. 29 Listopada 54, 31-425 Kraków, Poland; 2Department of Animal Molecular Biology, National Research Institute of Animal Production, Krakowska 1, 32-083 Balice, Poland; adrianna.musial@izoo.krakow.pl (A.D.M.); katarzyna.piorkowska@izoo.krakow.pl (K.P.); 3Department of Animal Reproduction, Anatomy and Genomics, University of Agriculture in Kraków, al. Mickiewicza 24/28, 30-059 Kraków, Poland; monika.stefaniuk-szmukier@urk.edu.pl

**Keywords:** Arabian horses, Thoroughbred, Polish Konik, draft horses, Hucul, genetic marker, peroxisomal acyl-coenzyme A oxidase 1

## Abstract

**Simple Summary:**

The genetic mechanisms occurring in organisms are shaped by selection pressure. Features that ought to be useful under given conditions leave their marks on the genome in the form of mutations, thereby creating different alleles. In this study, five different horse breeds were examined to find the connection between an individual’s lifestyle and the presence of the peroxisomal acyl-coenzyme A oxidase 1 (*ACOX1*) gene, which is necessary for some metabolic pathways. Results indicated that different *ACOX1* gene alleles play various roles in primitive breeds and domesticated horses. This led to the conclusion that the DNA profile can be rated on the basis of adaptation to living conditions, opening the gate for further investigation.

**Abstract:**

The *ACOX1* gene encodes peroxisomal acyl-coenzyme A oxidase 1, the first enzyme in the fatty acid β-oxidation pathway, which could be significant for organisms exposed to long periods of starvation and harsh living conditions. We hypothesized that variations within *ACOX1*, revealed by RNA Sequencing (RNA-Seq), might be based on adaptation to living conditions and had resulted from selection pressure. There were five different horse breeds used in this study, representing various utility types: Arabian, Thoroughbred, Polish Konik, draft horses, and Hucul. The single-nucleotide polymorphism (SNP) located in the *ACOX1* (rs782885985) was used as a marker and was identified using the PCR restriction fragment length polymorphism method (PCR-RFLP). Results indicated extremely different genotype and allele distributions of the *ACOX1* gene across breeds. A predominance of the G allele was exhibited in horses that had adapted to difficult environmental conditions, namely, Polish Konik and Huculs, which are considered to be primitive breeds. The prevalence of the T allele in Thoroughbreds indicated that *ACOX1* is significant in energy metabolism during flat racing.

## 1. Introduction

In the last 10 years, several studies were published that searched for the genetic background of different equine adaptations to effort. These adaptations are predominantly a result of selection pressure impacting the natural ability of horses. Since domestication, humans have shaped horses according to their own needs and in line with their natural abilities with respect to the environment, resulting in the creation of over 400 breeds [1]. In recent years, next-generation sequencing studies revealed the main targets of domestication and horse-breed differentiation at the genomic level [2]. Several genomic regions were indicated to be involved in the molecular processes associated, among other things, with skeletal muscle (organization, differentiation, and contraction), cardiac and skeletal systems, energy demands, brain development, and lipid deposition [3,4].

The transcriptome profiling of muscle tissue under training regimes has made it possible to detect molecular pathways with a strong potential connection to adaptation to effort. Under training conditions, fatty acid metabolism and fatty acid beta-oxidation were detected among the significantly deregulated networks. Moreover, the differential expression of the *ACOX1* gene (peroxisomal acyl-coenzyme A oxidase 1), a critical enzyme involved in beta-oxidation processes, was reported [5]. Due to the involvement of the *ACOX1* gene in lipid metabolism, it plays a key role during exercise and can also be of great importance in nutritional regulation [6]. The proper expression of the protein encoded by *ACOX1* appears fundamental in the metabolism of fat tissue, which is highly important for organisms exposed to long-term starvation. Therefore, it is of interest to investigate the *ACOX1* gene in terms of a possible connection to phenotypic variations observed across horse breeds.

The main purpose of the present study was to examine variation within the *ACOX1* gene in different horse breeds with various geographical origins and utility, Moreover, each breed represents a different kind of selection pressure. Arabian horses originated from the Arabian Peninsula, and were selected mainly for their exceptional strength and stamina in harsh desert conditions. Thoroughbreds originated in the British Islands over 400 years ago and are superior athletes with extraordinary racecourse performance. Recent studies indicated that the breed emerged as a cross between local mares, with the vast majority being British and Irish, and three founder stallions of oriental origin, including Turkoman and Barb stallions [7]. The Huculs and Polish Konik represent primitive breeds, described as wild or half-wild, possessing close to feral nutrition habits. The Konik are described as Tarpan relics [8], while Huculs originate from the Carpathian Mountains and are adapted to this kind of environment [9].

It is well established that horse breeds are differentiated not only with respect to external appearance, but also in terms of metabolism, energy demand, and proportions of muscle cells [10]. The presence of variations within the horse genome can be strongly connected with the horse’s nutrition strategy and living conditions. This is visible in a variety of different horse breeds [11]. For example, Polish draft horses were established after World War II as the crossbred of cold-blood mares and imported stallions from Ardennais and Breton stock [12]. The frequencies of alleles in general cases are considered to reflect the Hardy–Weinberg equilibrium, with any exceptions to this rule being forced by evolutionary influences such as selection pressure.

Thus, the main purpose of the present report was to examine the frequencies of the *ACOX1* genotypes in groups of five different horse breeds, namely, Arabian, Polish Konik, Hucul, Thoroughbred, and Polish draft horses, possessing different phenotypes and utilities. Each of these breeds presented different selection pressures, which was significant for finding the connection between their features and *ACOX1* allele frequencies.

## 2. Materials and Methods

### 2.1. Animals

A total of 615 horses were used in the study: Arabian, *n* = 290; Polish Konik, *n* = 94; Hucul, *n* = 67; Polish draft, *n* = 70; and Thoroughbred (TB), *n* = 94 (Appendix A). Biological material (hair follicles from 301 horses and whole blood from 314 horses) was sampled as a part of another experiment and stored in the Biological Material Bank of the National Research Institute of Animal Production and the Department of Horse Breeding, Institute of Animal Science, University of Agriculture, Krakow, Poland, in accordance with the requirements (Ethical Agreement no. 00665 and 1173/2015). Genomic DNA was isolated from whole blood and hair follicles using a Sherlock AX (A&A Biotechnology, Gdynia, Poland) according to protocol. DNA quantity and quality were checked using a NanoDrop 2000 spectrophotometer (ThermoFisher Scientific, Walham, MA, USA) and stored at −20 °C until further analysis.

### 2.2. Single-Nucleotide Polymorphism (SNP) Identification and Genotyping

SNP was identified using variant-calling analysis based on transcriptome data, including 10 individuals generated using RNA sequencing, as described by Ropka-Molik et al. [5]. In the first step, the raw quality of the reads was assessed using FastQC software v. 0.11.9 [13]. The reads were filtered using Flexbar software v.2.2 [14] to remove adapters and reads of phreak quality under 30. Minimal read length was set to 35. The next step was the mapping procedure, which was performed with the use of Tophat software v.2.1.1 [15], followed by the application of read groups to individual samples using Picard software [16]. The variant-calling procedure was performed using Freebayes software v1.3.1. After obtaining the raw VCF file, a series of filtration steps were applied using VCFtools v.4.2 to avoid false-positive results (total depth above 5 (DP > 5)), genotype quality >30, and sliding-window filtering of SNPs (maximum 3 SNPs in 35 bp window) [17].

The presence of polymorphism in the *ACOX1* locus was confirmed using Sanger sequencing. The primers were designed using Primer3 Input (version 0.4.0) on the basis of the ENSECAG00000022905 reference (Table 1), and the PCR amplicon was obtained using AmpliTaq Gold 360 Master Mix (Thermo Fisher Scientific) according to protocol. The purified PCR products (Eppic reagent; A&A Biotechnology) were used for Sanger sequencing performed using the BigDye Therminator v3.1 Cycle Sequencing Kit and BigDye XTerminator Purification Kit (Thermo Fisher Scientific). Capillary electrophoresis was carried out on a 3500xL Genetic Analyzer (Applied Biosystems; Thermo Fisher Scientific).

The PCR restriction fragment length polymorphism (PCR-RFLP) method was designed for the selected polymorphisms in the *ACOX1* gene, with specific endonucleases recognizing particular gene variants (NebCutter v.2.0). PCR reaction was performed using an AmpliTaq Gold 360 Master Mix (Thermo Fisher Scientific) according to protocol. Primer sequences are shown in Table 1. Then, PCR products were digested overnight at 37 °C with *DdeI* endonuclease and separated in 3% agarose gel.

The significance of genotype distribution among the analyzed horse breeds was calculated using the chi-squared test [18].

## 3. Results

Genotyping analysis of alleles across five horse breeds indicated a few significant differences among the breeds. The highest percentages of the TG (c.238TG) genotype were detected in Polish draft horses and TB populations (47.1% and 43.6, respectively), and the lowest TT (c.238TT) individuals occurred in Arabian horses and Polish Konik. In the Hucul breed, the TT genotype was not present, and GG (c.238GG) horses accounted for 97%. Genotype distribution was significantly distinct among the investigated horse breeds (Table 2). The highest number of TT individuals was identified in the TB breed (41.5%), a result that was significant compared to those of the other breeds (Table 3 and Figure 1). The TB population exhibited similar frequency for both the TG and the TT genotype (43.6% and 42.5%, respectively), while the Hucul breed was almost monomorphic. Moreover, all investigated horse populations were consistent with the Hardy–Weinberg equilibrium (Table 3 and Figure 1).

## 4. Discussion

The *ACOX1* gene encodes peroxisomal acyl-coenzyme A oxidase 1, an enzyme of fatty acid beta-oxidation, which catalyzes the desaturation of acyl-Coenzyme A to 2-trans-enoyl-CoA [19]. ACOX oxidase catalyzes the first step in the fatty acid beta-oxidation pathway and donates electrons to molecular oxygen, producing hydrogen peroxide CoA (Appendix A). The role of this pathway, which occurs in mitochondria and peroxisomes, is to shorten the long-, medium-, and short-chain fatty acids derived from the diet via the oxidation of fatty acids to generate energy [20]. In humans, mutations in the *ACOX1* gene may cause pseudo-neo-natal adrenoleukodystrophy, a disease that causes the accumulation of long-chain fatty acids, resulting in the demyelination and impairment of nervous signal transmission [21]. Moreover, alternatively spliced transcript variants encoding different isoforms related to adrenoleukodystrophy were identified [22]. In horses, the reduction of acyl-CoA dehydrogenase activity is referred to as equine atypical myopathy and is caused by hypoglycin A intoxication associated with the ingestion of sycamore maple tree seeds. Hypoglycin A, metabolized to inhibit acyl-CoA dehydrogenase, results in the accumulation of acyl-CoAs in the mitochondria, leading to multiple acyl-CoA dehydrogenase deficiency (MADD) [23]. Furthermore, faults with fatty acid metabolism were recognized in many cases of exertional rhabdomyolysis [24]. Moreover, very long chain fatty acids (>C20) are mostly beta-oxidized in the peroxisomes, and this pathway is catalyzed by AOX, L-PBE (Peroxisomal Enzyme L-PBE), and thiolase, which are transcriptionally activated by Peroxisome proliferator-activated receptor—PPARα (Appendix A) [25]. This interaction of *ACOX1* and PPARα might be regulated by the presence of oleoylethanolamide (OEA) [26,27], which is the major regulator of satiety by peripheral regulation of feeding [28]. Furthermore, OEA enhances β-adrenergic receptor activation, which promotes brown adipose tissue β-oxidation and thermogenesis [26].

Feral populations of herbivores can reduce their metabolism and energy needs under harsh environmental conditions, including low temperatures and limited food availability [29]. In feral and semiferal equine populations, dry-matter intake (DMI) varies during the year, with higher values during the summer and autumn, and lower values in winter and early spring. The lower DMI during winter also indicates to use body fat reserves to fuel metabolism and shift to catabolic metabolism [30]. Furthermore, primitive semiferal Shetland ponies can reduce their seasonal activity, similarly to Przewalski horses. They acclimatize to difficult winter conditions by hypothermia, changing their metabolic rate and physiological parameters (hypometabolism) [31].

In our study, we compared the distribution of g.6105340T>G single nucleotide polymorphism within the coding region of the *ACOX1* gene, which changes serine to alanine at position 80 of the amino acid sequence, across five different horse breeds. The investigated polymorphism causes amino acid substitution in the N-terminal domain of acyl-coenzyme A oxidase, a domain that is responsible for the oxidase function containing the expected fatty acyl binding site [32]. Therefore, any amino acid modifications within this part may affect the proper enzymatic function of the *ACOX1* enzyme by disrupting the substrate-binding process. Analysis of the g.6105340T>G polymorphism across different horse breeds showed that a higher frequency of the G allele was observed in breeds in environmental conditions with limited food availability. In the Hucul population, the TT genotype was not identified, and TG heterozygotes only accounted for 3%. The predominance of the G allele in Arabian horses, adapted to desert conditions; Polish Konik, perfectly suited to European forests; and Hucul Carpathian horses, adjusted to mountain conditions, suggests that the G allele might not be under past selection pressure.. The assembly of the equine reference genome is based on individually represented Thoroughbred horses. The TB breed was created in the British Islands under intense selection pressure for racing purposes, with the provision of excellent conditions for herbivores; thus, adaptation to long-term starvation was useless [33]. In turn, Hucul horses are considered to be a primitive breed [34], with their habitat being cold mountain areas [35]. Therefore, they are strongly adapted to harsh living conditions and long-term starvation. Their most important ability is building suitable fat tissue within short periods of time while having a primitive nutrition strategy. In Huculs, during winter, corresponding to difficult living conditions, fatty acid metabolism is very efficient, which perhaps comes from natural selection pressure.

The present results demonstrated the opposite distribution of *ACOX1* genotypes in TB compared to Huculs and Polish Konik. PPARα-targeted genes are the main factors controlling lipid metabolism and the maintenance of energy homeostasis [36,37,38]. In horses, genes belonging to the PPAR family are exercise-regulated and potential molecular factors affecting exercise performance [39,40]. Furthermore, PPARGC1A, a direct regulator of PPARα, is under intense selection pressure due to the energy metabolism requirements in flat racing [33,39,40]. Since there is evidence that *ACOX1* is related to feeding control by PPARα regulation [27], it can be hypothesized that selection pressure toward the speed phenotype in racing horses might also affect *ACOX1* gene-encoded peroxisomal straight-chain Acyl-CoA oxidase. Furthermore, in a recent study by Fontanel et al. [41], the *ACOX1* gene was shown to be significantly associated with racing performance in Arabian horses, suggesting that *ACOX1* might be under selection for energetic efficiency under difficult conditions. While TB horses are under selection towards speed performance, the selection of Arabian horse can differ depend on population and usability type. The French population of Arabians are mainly racing horses, whereas Polish Arabian horses, studied in this report, belong to the Polish population known to be under selection for proper conformation for endurance with strong ability to race. The previous report [41] indicated an association of the *ACOX1* T allele with flat racing performance in Arabian horses. On the other hand, the frequency of TT horses limited the strength of obtained concern. The selection pressure in TB breed and the frequency results showing that TB horses were characterized by the highest rates of TT genotype (41.5%) and T allele (63%) can support the conclusion that T variant affected flat racing abilities.

In turn, in cold-blood horses, selection pressure was similarly directed toward power. Our results made it possible to identify a significantly different genotype distribution in Polish draft horses compared to that in other breeds, demonstrating advantages in terms of both heterozygote horses (47.14%) and GG horses (40%). The most recent study performed on Arabian horses showed significant association between the *ACOX1* genotype and the financial benefits accruing from winnings (earnings and earnings per start) in flat racing [41]. The authors indicated that this gene could be considered a genetic marker related to gallop-racing performance traits and pointed to the need for further research in this area.

## 5. Conclusions

In the present report, we detected the extremely different genotype and allele distributions of the *ACOX1* gene across horse breeds corresponding to various utility types. Acyl-coenzyme A oxidase 1 is probably a strategic metabolic enzyme responsible for lipid metabolism, resulting from adaptation to living conditions and the type of performed physical exertion. Our results indicated a predominance of the G allele in horse breeds that are very well adapted for existing in difficult environmental conditions (mountains, forests, deserts), which may suggest that this allele was not under past selection pressure. Moreover, the high number of TT horses and the prevalence of the T allele in TB indicated that *ACOX1*, together with other genes involved in PPAR signaling pathways, will be under selection pressure due to the energy metabolism necessary in flat racing. The presented results indicate the need for further research on the *ACOX1* gene with respect to its relationship with performance features in racing horses.

## Figures and Tables

**Figure 1 animals-10-02225-f001:**
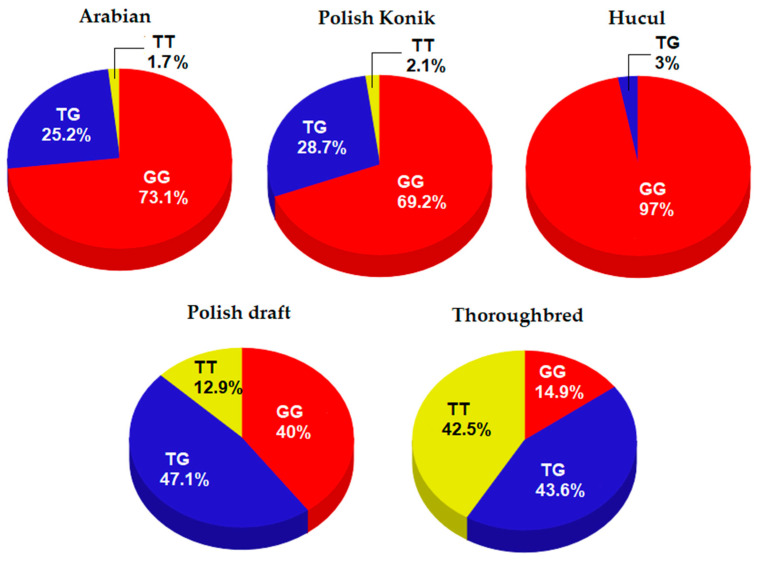
Genotype distribution as percentage across analyzed horse breeds.

**Table 1 animals-10-02225-t001:** Parameters of used method and investigated polymorphism features.

Gene	*ACOX1*
Accession number	ENSECAG00000022905
SNP	rs782885985
	NC_009154.3:g.6105340T>G
	ENSECAT00000024965.2:c.238T>G
	ENSECAP00000020757.2:p.Ser80Ala
Primers	**F**: CAGCTGTGATTACGGGAGGT
	**R**: TGAAAACGTGCAGTTTGAGC
PCR-RFLP (PCR restriction fragment length polymorphism) conditions	*DdeI* endonuclease
	Alleles: T–197, 145 bp; G–342 bp

**Table 2 animals-10-02225-t002:** Differences in genotype distribution among analyzed breeds according to the chi-squared test.

	Arabian	Polish Konik	Hucul	Polish Draft	Thoroughbred
Arabian horses		ns	0.0001	<0.0001	<0.0001
Polish Konik	ns		0.00005	0.0002	<0.0001
Hucul	0.0001	0.00005		<0.0001	<0.0001
Polish draft horses	<0.0001	0.0002	<0.0001		0.00002
Thoroughbred	<0.0001	<0.0001	<0.0001	0.00002	

ns—not significant.

**Table 3 animals-10-02225-t003:** Allele frequency and percentage distribution of genotypes across breeds: significance according to Hardy–Weinberg equilibrium (HWE).

	Arabian			Polish Konik			Hucul			Polish Draft			Thoroughbred		
Genotype	GG	TG	TT	GG	TG	TT	GG	TG	TT	GG	TG	TT	GG	TG	TT
Number of horses	212	73	5	65	27	2	65	2	0	28	33	9	14	41	39
Percentage of genotypes (%)	73.1	25.2	1.7	69.2	28.7	2.1	97.0	3.0	0.0	40.0	47.1	12.9	14.9	43.6	42.5
HWE significance	0.65			0.67			0.90			0.08			0.55		
Minor allele frequency	0.14			0.16			0.01			0.30			0.63		

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
