# Peer review of "Variability of ACOX1 Gene Polymorphisms across Different Horse Breeds with Regard to Selection Pressure"

_animals, 2020, doi:10.3390/ani10122225_

Round 1

Reviewer 1 Report

Dear Authors

The manuscript quality has greatly improved after english revision. Just a comment. Table 3 and Figure 1 report the same information. Please remove the line in the table bringing the % genotype or in alternative remove the figure. moreover 41.5% TT for Thoroughbred should be 42.5%.

Reviewer 2 Report

This manuscript is interesting and the genotyping procedure with hundreds of horses is appropriate.

The value of this manuscript and data is high and this reviewer thinks it needs to be published without much delay. 

I wonder if they could simply look up other published horse breeds' ACOX1 genotypes easily (like Mongolian and Korean horses).

>The presented 236 results indicate the need for further research on the >ACOX1 gene with respect to its relationship with
>237 performance features in racing horses.

ACOX1 is an important gene and I think this suggestion is fair.

>will be under selection pressure due to the energy metabolism

>necessary in flat racing.

This is a speculation based on statistically prominent frequency differences.

Minor:

>which may suggest that this allele was not
>233 under past selection pressure..

There are two periods.

Reviewer 3 Report

Dear Authors,

The study is interesting, have a basis for new studies and discoveries. I propose the study for publication, after minor corrections. Congratulations on the work presented.

Line 22 - The word “conclusion” may be a little strong.

Line 41 - Refer to some of the studies.

Lines 60 and 79, and the following sentences of each line - These are repetitions of the aims of the study, you need to cut lines 60 and leave the last paragraph of the Introduction to write the aims.

Table 3 Percentage of genotypes (%) - The results are already shown in Figure 1, they do not need to be repeated, they can be eliminated in Table 3.

Conclusion - The present many results / discussion, you can write this part in the last paragraph of the discussion and present a conclusion more directly.

Supplementary material – The photo in the file “Sup file 2 Thoroughbred (photo credit BogusÅ‚awa DÅ‚ugosz all rights reserved)” does not correspond to a Thoroughbred horse, needs to be changed. Take advantage and change the photo in the file “Sup file 1 The Arabian horse (photo credit Ewa Imielska-Hebda all rights reserved)” to a position of the horse identical to the other photos. It does not seem logical to have a horse in motion and the others to be in a standing lateral position (better to observe the differences between the races).